# Supply-side readiness to deliver HIV testing and treatment services in Indonesia: Going the last mile to eliminate mother-to-child transmission of HIV

**Rabiah al Adawiyah** [1]*, **David Boettiger** [1], **Tanya L. Applegate**[1], **Ari Probandari** [2], **Tiara Marthias** [3], **Rebecca Guy**[1], **Virginia Wiseman**[1,4]

**1** The Kirby Institute, University New South Wales, Sydney, Australia, **2** The Faculty of Medicine, Universitas Sebelas Maret, Surakarta, Indonesia, **3** Nossal Institute for Global Health, The University of Melbourne, Melbourne, Australia, **4** The London School of Hygiene and Tropical Medicine, London, United Kingdom

* radawiyah@kirby.unsw.edu.au

## Abstract

### Introduction

Despite national efforts to integrate Prevention of Mother-to-Child Transmission (PMTCT) of HIV services into antenatal care in Indonesia, the rate of mother-to-child transmission of HIV remains the highest in the world. A range of barriers to uptake and long-term engagement in care have been identified, but far less is known about health system preparedness to deliver PMTCT of HIV services. This study explored supply-side barriers to the delivery of PMTCT services in Indonesia and whether these factors are associated with the uptake of antenatal HIV testing.

### Materials and methods

An ecological analysis was undertaken, linking data from the World Bank Quality Service and Delivery Survey (2016) with routine data from Indonesia's HIV and AIDS case surveillance system and district health profile reports (2016). Supply-side readiness scores—generated from a readiness index that measures overall structural capacity and is often used as proxy for quality of care—were adapted from the WHO Service Availability and Readiness Assessment and presented by sector and geographic area. Univariate and multivariate regression analysis was used to explore factors associated with the uptake of antenatal HIV testing in public facilities.

### Results

In general, public facilities scored more highly in most inputs compared to private facilities. Facilities located in urban areas also scored more highly in the majority of inputs compared to ones in rural areas. Readiness scores were lowest for PMTCT services compared to Antenatal Care and HIV Care and Support services, especially for the availability of medicines such as zidovudine and nevirapine. The national composite readiness score for

**Funding:** Corresponding author, RA, received PhD funding from the University of New South Wales through the Scientia PhD scholarship scheme. The funders had no role in study design, data collection and analysis, decision to publish, or preparation of the manuscript.

**Competing interests:** The authors have declared that no competing interests exist.

PMTCT was only 0.13 (based on a maximum score of 1) with a composite score of 0.21 for public facilities and 0.06 for private facilities. The multivariate analysis shows that the proportion of pregnant women tested for HIV was more likely to be greater than or equal to 10% in facilities with a higher readiness score and a higher number of trained counsellors available, and less likely in facilities located outside of Java-Bali and in facilities supporting a higher number of village midwives.

## Discussion

Despite targeted efforts by the Indonesian government and multinational agencies, significant gaps exist in the delivery of PMTCT that compromise the standard of care delivered in Indonesia. Future strategies should focus on improving the availability of tests and treatment, especially in the private sector and in rural areas.

## Introduction

Mother-to-child transmission, which is also known as 'vertical transmission', is the most common cause of HIV infection in children under 15 years of age [1]. In the absence of treatment, the risk of mother-to-child transmission of HIV is 15% to 30% during gestation or labour, with an additional transmission risk of 10% to 20% associated with prolonged breastfeeding [2]. However, antiretroviral treatment (ART) and other interventions can reduce this risk to below 5% [2]. In 2015, more than 1.4 million pregnant women were infected with HIV, and mother-to-child transmission of HIV led to over 150 000 infant cases globally due to gaps in access to diagnosis and treatment [3]. Only around 53% of the 1.8 million children living with HIV were receiving ART in 2019, and among those without access to effective treatment, up to 50% died due to AIDS-related illnesses [4]. Elimination of mother-to-child transmission of HIV has therefore been identified as a global public health priority [5].

Transmission of HIV infection from mother-to-child, can be prevented with the use of affordable and reliable rapid antibody tests to support early HIV diagnosis and treatment in pregnancy. The low cost and simplicity of these tests and the immediacy of results greatly benefit resource-constrained settings by allowing the rapid initiation of treatment [6, 7]. In 2010, the World Health Organization (WHO) published the Prevention of Mother-to-Child Transmission (PMTCT) guidelines which were later updated to include recommendations for lifelong ART for all HIV-positive pregnant and breastfeeding women regardless of their CD4 count or clinical stage (i.e., Option B+) [8]. The updated PMTCT strategy is designed to: achieve the elimination of new paediatric HIV infections; ensure all ART-eligible pregnant women receive treatment; and contribute to ending the AIDS epidemic by 2030 [9–11]. Testing and treatment targets have been set by the WHO, which recommend countries to aim for 95% coverage of antenatal care (ANC), 95% HIV testing coverage among pregnant women, and 95% treatment coverage for women diagnosed and living with HIV (known as the WHO 95/95/95 target) [12].

In 2005, Indonesia adopted PMTCT as a national policy, following a voluntary counselling and testing approach [13, 14]. Despite these guidelines and the high coverage of ANC services across the country, only an estimated 27% of pregnant women were tested for HIV in 2017 [15]. In 2016, mother-to-child transmission risk in Indonesia was estimated at 27%, making it the highest-ranking country among the 23 countries with the highest numbers of new HIV

infections in children, adolescents and young women [16]. In 2016, Indonesia was also ranked 7th among countries with the highest number of new HIV paediatric infections with an estimated 4 950 new infections that year [15]. In 2017, the national PMTCT guidelines were updated to integrate HIV services into maternal care *("Standar Pelayanan Minimal/SPM")*, making HIV testing a part of routine screening for pregnant women [17, 18]. This led to an expansion in HIV testing among pregnant women, with coverage increasing from 27% in 2017 to 45% in 2019 [15, 17]. Disappointingly, less than half of those women testing positive received HIV treatment in 2019 [19].

The low antenatal HIV testing and treatment coverage in Indonesia poses a major challenge to reaching the WHO 95/95/95 targets and the Government's target of triple elimination of HIV, syphilis and hepatitis B by 2022 [12]. Studies in low-and middle-income countries (LMICs) reveal a range of individual factors affecting the uptake of testing and treatment for HIV by pregnant women including age, socioeconomic status, distance to the health facility, education level, marital status, stigma, health knowledge, previous pregnancy complications and the number of ANC visits [20, 21]. Far fewer studies have examined the influence of supply-side factors on the delivery of PMTCT services, especially in the Asia-Pacific region [22]. Service availability and readiness, which reflects the system level willingness and preparedness to provide quality services [23], has been shown to be lacking in several key areas of ANC delivery in Indonesia including diagnostic capacity and medicine supply [15, 24, 25]. This study will be the first in Indonesia to assess the readiness and availability of both public and private health services to deliver PMTCT services, a key element of ANC, and to examine the role of supply-side factors in the uptake of antenatal testing for HIV.

## Materials and methods

### Study design

We undertook an ecological study based on survey data and routinely collected administration data at the health facility level to investigate variations in supply-side readiness (by geographic area, sector, and facility type) and associations between supply-side readiness and HIV testing coverage.

### Setting

Indonesia is a lower-middle-income country comprised of over 15 000 islands, 34 provinces and 416 districts [26]. It is the fourth largest country in the world with a population of more than 275 million people, of which 56% live in urban areas [1]. Indonesia has approximately 5.3 million annual births and a coverage rate of around 95% for first ANC attendance [15]. It has a national HIV prevalence of around 0.3% in pregnant women, except for a few provinces such as Papua where HIV prevalence is close to the 1% [15]. Maternal healthcare services in Indonesia, which include PMTCT services, are provided through a complex network of public and private providers. Indonesia's primary health centres or *"puskesmas"* form the backbone of the public health system, with a network of almost 10 000 facilities across the country [27]. *"Puskesmas"* are linked to a wider network of auxiliary health centres, called *"pustu"* that provide community outreach services. *"Puskesmas"* also supervise and support a network of smaller community-based units including *"polindes"* (i.e., village maternity clinics), *"poskesdes"* (i.e., village health posts) and "*posyandu*" (i.e., integrated community-based health service posts). Private maternal health providers include private hospitals, multi-practitioner private maternity clinics ("*klinik bersalin*") and single-practitioner home-based or clinic-based midwives. In 2013, more than 52% of pregnant women sought care at private midwife clinics, 16.6% at *"puskesmas"*, 10% at hospital, 10% at *"posyandu"*, 6% at *"poskesdes"* and 4.3% at private

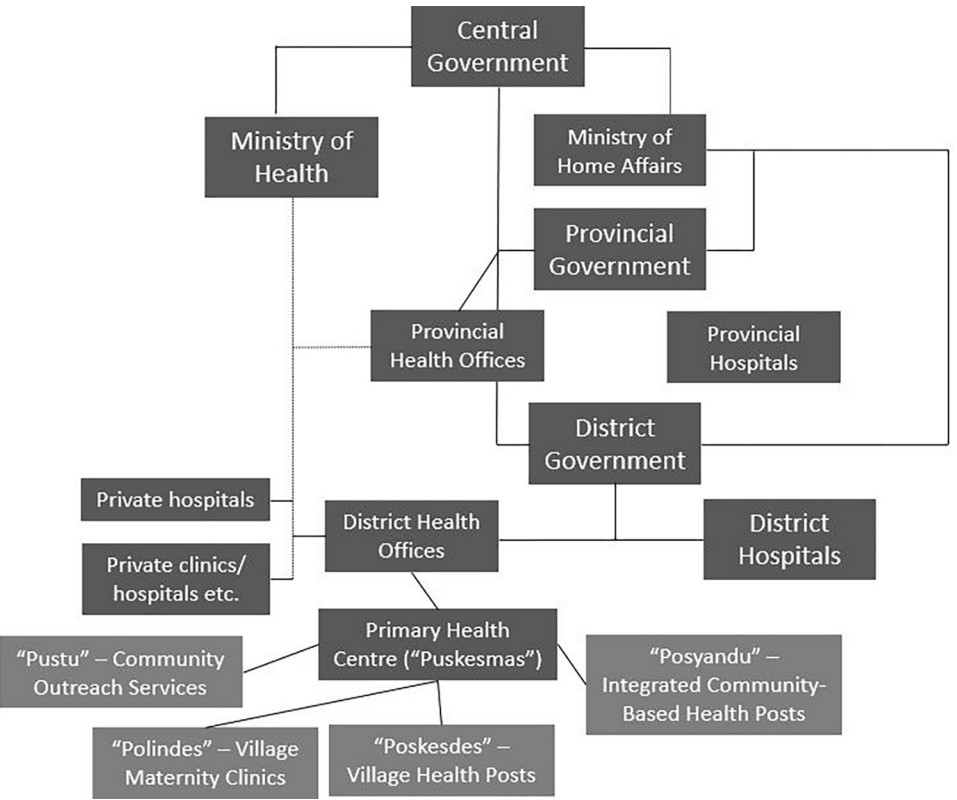

**Fig 1. Organization of health system in Indonesia.**

obstetrician/general practitioner clinics [28]. Village midwives are the dominant provider of maternal health care and have been dramatically scaled up by the Government of Indonesia since 1984 [29]. Some village midwives are employed by the central or local government, however, most of them have multiple sources of income, with an estimated 58% of their annual total income coming from private practices [30]. In 2014, Indonesia also launched a comprehensive national social health insurance scheme known as the "*Jaminan Kesehatan Nasional*" or simply referred to as the JKN. This scheme represents the biggest single-payer health insurance system in the world—covering around 225 million beneficiaries in 2021 [31–33]. Under the JKN, the majority of *"puskesmas"* revenue comes from capitation payments for JKN members [34]. Around 42% of private clinics, 60% of private hospitals and 14% of general practitioners were empanelled with the JKN in 2015 (34). Private midwife practices operate outside of the JKN and patients must pay out-of-pocket for the services [35]. Fig 1 summarised the organization of health system in decentralized context of Indonesia [36].

### Data sources

This study used three linked secondary datasets:

1. Quality Service and Delivery Survey (QSDS), 2016: This survey was undertaken by the World Bank Indonesia team in consultation with the Indonesian Ministry of Health (MoH) to assess the supply-side readiness across public and private primary care facilities in Indonesia [37]. For this study, we used the QSDS national sample dataset, based on a stratified, multistage, cluster sampling frame designed to provide a sample that is representative at the national level. There were two strata—cities and districts—from which a sample of 22

cities/districts was drawn from a total of 514 using simple random sampling with equal probability of selection. At the second stage of sample selection, health facilities in each chosen district/city were stratified by public primary care or private primary care (polyclinics and private medical practices). Samples of predetermined size were then chosen from each stratum via simple random sampling with equal probability of sampling, regardless of the catchment population, utilisation, or the size of the health facility (in terms of staff or financing). A total of 268 public primary care facilities and 289 private medical practices (single general practitioner or multiple general practitioners' clinics) were selected from 915 public primary care facilities and 835 private medical practices. A detailed description of the sampling methodology for the QSDS can be found elsewhere [37].

2. National System for Information on HIV/AIDS (SIHA): The number of pregnant women tested for HIV (i.e., at least one HIV rapid antibody test) in public primary care facilities from 2016 to 2018 were provided to our team by the Indonesian MoH, sourced from the SIHA database. SIHA is a web-based routine surveillance system for HIV/AIDS and Sexually Transmitted Infections. It operates at the district, provincial and national levels and is updated quarterly. For this study, the SIHA database provided information from the public and private providers on the number of pregnant women tested, the number of pregnant women with a positive result, and the number of pregnant women testing positive that initiate treatment.

3. District Health Office (DHO) Annual Reports: Data on the number of pregnant women attending antenatal clinics in each facility included the QSDS was sourced from DHO annual reports. These data are publicly accessible through district government websites [38–44].

### Study outcomes

**Proportion of pregnant women tested for HIV per facility.**   The proportion of pregnant women tested for HIV per facility was defined as the total number of pregnant women per year tested for HIV per facility, divided by the total annual number of pregnant women attending their first antenatal visit per facility. For the regression model, the primary outcome was defined as testing for HIV among pregnant women per facility greater than or equal to 10% ($\geq$10%)–this cut-off represented the national testing coverage for HIV among pregnant women in Indonesia in 2016 [15].

**Explanatory variables.**   A conceptual framework, developed by Jacobs et al. (2012) to categorise different barriers to accessing health services in low-income Asian countries, was used to guide the selection of explanatory variables for our model of HIV testing among pregnant women [45]. Following the Jacobs et al. approach, eight variables were grouped under four core themes: geographical access, availability, affordability, and acceptability. Explanatory variables included: health facility readiness score, geographical location (i.e., urban/rural and Java-Bali/Outer Java-Bali), type of financial management structure (i.e., for-profit known as *"Badan Layanan Umum Daerah/BLUD"* or not-for-profit *"non-BLUD"*) and several variables related to outreach activities such as the number of village midwives, active community health posts (*"posyandu"*), active community health workers (CHWs), and the availability of counsellors at each facility. The categorisation of Java-Bali and Outer Java-Bali is based on two major geographic regions in Indonesia, with Outer Java-Bali covering all other islands spread sparsely across the country. Java-Bali is the most densely populated region containing around 60% of the total population and has a stronger health system supply-side compared to Outer Java-Bali.

Systematic reviews show that these variables are commonly associated with the effective delivery and uptake of HIV testing and treatment in ANC in resource-constrained settings [20, 25, 46]. The full list of variables and their definitions are provided in S1 Table.

Since this is an ecological study using data collected at the health facility level, supply-side readiness will from hereon be referred to as 'health facility readiness'. The health facility readiness scores were calculated for ANC, PMTCT, and HIV care and support services (HCS). These three readiness scores (i.e., for ANC, PMTCT and HCS) represent the different steps in the PMTCT program cascade of care. Indicators of health facility readiness were adapted from the service-specific readiness indicators included in the WHO Service Availability and Readiness Assessment (SARA) tool [47]. These indicators reflect the ability of a health care facility to provide specific services including staff and training, equipment, diagnostics, medicines and commodities [47, 48]. There is no specific readiness threshold, however, this metric is important to indicate the capacity of facilities to provide specific services as well as identify within- and between- countries' differences [49]. For each indicator, a binary variable was created. The indicator was coded as '1' if the facility met the criteria and '0' if it did not. The composite readiness score represents the sum of total service readiness scores divided by the total number of indicators. The composite readiness score is calculated for ANC, PMTCT and HCS to provide descriptive information on health facility readiness for each cascade of care along with the PMTCT program.

**Statistical analysis.** For this study, the World Bank QSDS data set was used to measure the availability and supply-side readiness by region and sector (public-private), while the SIHA and DHO reports were linked to the QSDS to explore the associations between the availability/readiness and uptake of HIV testing by pregnant women through univariate and multivariate analysis. A total of 268 public and 289 private facilities in the QSDS dataset were matched with the annual facility-level data from the SIHA database in 2016 and the DHO reports. The health facility (at the primary care level) is the unit of analysis.

Descriptive analyses included the number and proportions of facilities offering PMTCT across districts, public-private providers, and between urban and rural areas in Indonesia. The indicators used for readiness scores of ANC, PMTCT and HCS were examined using Cronbach's alpha, a popular method to measure reliability of a score [50]. The ANC readiness indicators had a Cronbach's alpha of 0.824, while the PMTCT and the HCS readiness indicators had a Cronbach's alpha of 0.630 and 0.717, respectively. This indicates an acceptable level of internal consistency of indicators used to construct the score [51].

Univariate and multivariate regression were used to explore the relationship between the explanatory variables and the uptake of HIV testing by pregnant women across three types of services (ANC, PMTCT and HCS) in public health facilities. For regression models, ANC, PMTCT and HCS the readiness score were transformed into categorical variables: for ANC, 0 referred to an ANC readiness score of 0 to 0.65, 1 referred to 0.66 to 0.80 and 2 referred to 0.81 to 1, for PMTCT, 0 referred to PMTCT readiness score of 0 to 0.22, 1 referred to 0.23 to 0.40 and 2 referred to 0.41 to 1, and for HCS, 0 referred to HCS readiness score of 0 to 0.35, 1 referred to 0.36 to 0.50 and 2 referred to 0.51 to 1. Models were based on weighted samples to correct for the differential probability of a district and facility being included in a sample. The primary outcome was defined as testing for HIV among pregnant women per facility greater than or equal to 10% ($\geq$10%)–a cut-off represented the national testing coverage for HIV among pregnant women in Indonesia in 2016 [15]. The proportion of pregnant women tested for HIV per facility was matched with the sampled facilities provided in the QSDS. Factors associated with the outcome of interest were first assessed using univariate logistic regression models. Covariates with an overall statistical significance of $p<0.1$ level in the univariate analysis were included in the final reduced multivariate models. We present the odds ratios and

95% confidence intervals (CIs). Sensitivity analysis was performed to test the impact of changing the threshold for the proportion of pregnant women tested for HIV per facility to 5% and 15% and to test the impact of using modified PMTCT readiness scores and provided in S2 and S3 Tables, respectively. All analyses were performed using STATA V16.1.

**Ethical approval.** The Human Research Ethics Advisory Panel at the University of New South Wales (HC180439) and the Health Research Ethics Committee in the Faculty of Medicine at the Universitas Sebelas Maret (205/UN27.6/KEPK/2018) approved the use of secondary data for this study. All data were fully anonymized before accessing it.

## Results

### Health facility availability and readiness

Our analyses were based on 268 public facilities ("*puskesmas*") and 289 private facilities (GP clinics, single and multi-practitioners) across 22 districts and 11 provinces in Indonesia. Table 1 presents the percentage of facilities with each type of indicator, across the service areas of ANC, PMTCT and HCS. Indicators of readiness were lowest for PMTCT services, with around 17% and 19% of facilities having guidelines and staff trained in PMTCT available, respectively. Further, only 21% and 5% of facilities had HIV rapid test kits and maternal antiretroviral prophylaxis available, respectively.

Fig 2 shows differences in the availability of inputs needed to deliver HCS, PMTCT and ANC across public/private facilities and urban/rural locations. In general, public facilities scored more highly in most of inputs (26 out of 27 indicators) compared to public facilities. Facilities located in urban areas scored more highly in majority of inputs (24 out of 27 indicators) compared to ones in rural areas. Trained staff for PMTCT were available at 47% and 12% of public facilities, in urban and rural areas, respectively, while HIV rapid test kits were available at 56% and 25% of public facilities in urban and rural areas, respectively. The availability of same indicators was considerably lower in private compared to public facilities, with trained staff for PMTCT available at 7% and 5% of facilities in urban and rural areas, respectively, while HIV rapid test kits were not available at either all. The only readiness indicator for which private facilities performed better than public facilities was the availability of visual and auditory privacy rooms (55% and 36% of private facilities in urban and rural areas, compared to 34% and 13% in public facilities in urban and rural areas).

Fig 3 presents the composite readiness scores for ANC, PMTCT and HCS services for public and private facilities in urban and rural areas. The composite readiness scores were 0.48 for ANC (range 0–1), 0.13 for PMTCT (range 0–0.78), and 0.30 for HCS (range 0–0.89). The composite readiness score for ANC services was the highest, with 0.79 and 0.71 for public facilities in urban and rural areas, respectively. Comparative scores were lower in private facilities, at 0.43 and 0.42 in urban and rural areas, respectively. The composite readiness scores for PMTCT were the lowest across all service types, with 0.26 and 0.13 for public facilities in urban and rural areas, respectively. Readiness scores were lower in private facilities, with 0.06 and 0.04 in urban and rural areas, respectively.

Fig 4 shows the composite readiness scores for ANC, PMTCT and HCS by district (public and private facilities combined). While the national composite readiness scores were 0.58 (range 0.37–0.75), 0.13 (range 0.08–0.16) and 0.31 (range 0.03–0.54), for ANC, PMTCT and HCS, respectively, there was substantial variation across districts. The composite readiness scores for ANC were the highest across all districts, followed by HCS and PMTCT. The lowest HCS and PMTCT readiness scores were recorded for the district of Banjar–ANC services in this district received a composite readiness score of 0.61 compared to 0.08 and 0.03 for HCS and PMTCT services, respectively.

**Table 1. Service readiness indicators for ANC, PMTCT and HCS in 2016.**

| Service area (total indicators) | Readiness indicator | Study facilities with indicators n (%) |
| --- | --- | --- |
| ANC (8 indicators) | Guidelines on ANC | 189 (33.9) |
| | Staff trained in ANC | 164 (29.4) |
| | Blood pressure apparatus | 525 (94.2) |
| | Haemoglobin | 292 (52.4) |
| | Urine dipstick protein | 243 (43.6) |
| | Iron tablets | 247 (44.3) |
| | Folic acid tablets | 262 (47.0) |
| | Tetanus Toxoid Vaccine | 225 (40.4) |
| | **Composite readiness score for ANC services** | **0.48** |
| PMTCT (14 indicators) | Guidelines for PMTCT | 94 (16.9) |
| | Guidelines for infants and young child feeding | 171 (30.7) |
| | Guidelines for antiretroviral therapy | 8 (1.4) |
| | Staff trained in PMTCT | 106 (19.0) |
| | Staff trained in infant and young child feeding | 125 (22.4) |
| | Visual and auditory privacy | 214 (38.4) |
| | HIV rapid testing kit (HIV diagnostic capacity for adults) | 116 (20.8) |
| | Zidovudine (AZT) | 5 (0.9) |
| | Nevirapine (NVP) | 6 (1.1) |
| | Maternal antiretroviral prophylaxis | 25 (4.5) |
| | CD4 or viral load | 7 (1.3) |
| | Renal function test | 69 (12.4) |
| | Liver function test | 61 (10.9) |
| | Three first line antiretroviral Option B+ (AZT + NVP + 3TC) | 6 (1.1) |
| | **Composite readiness score for PMTCT services** | **0.13** |
| HIV care and support (HCS) (9 indicators) | Guidelines for clinical management of HIV/AIDS | 1 (0.2) |
| | Guidelines for management of HIV and TB co-infection | 81 (14.5) |
| | Staff trained for clinical management of HIV/AIDS | 20 (3.6) |
| | Training on management of HIV and TB co-infection | 133 (23.9) |
| | System for diagnosis of TB among PLHIV | 40 (7.2) |
| | Intravenous solution with infusion set | 346 (62.1) |
| | Co-trimoxazole | 407 (73.1) |
| | First-line TB treatment medication | 311 (55.8) |
| | Condoms | 193 (34.5) |
| | **Composite readiness score for HCS services** | **0.30** |

3TC: Lamivudine; AIDS: Autoimmune Deficiency Syndrome; ANC: Antenatal Care; AZT: Zidovudine; CD4: Cluster of Differentiation 4; Human Immunodeficiency virus; LMV: Lamivudine; NVP: Nevirapine; PLHIV: People Living with HIV/AIDS; PMTCT: Prevention of Mother-to-Child Transmission; TB: Tuberculosis

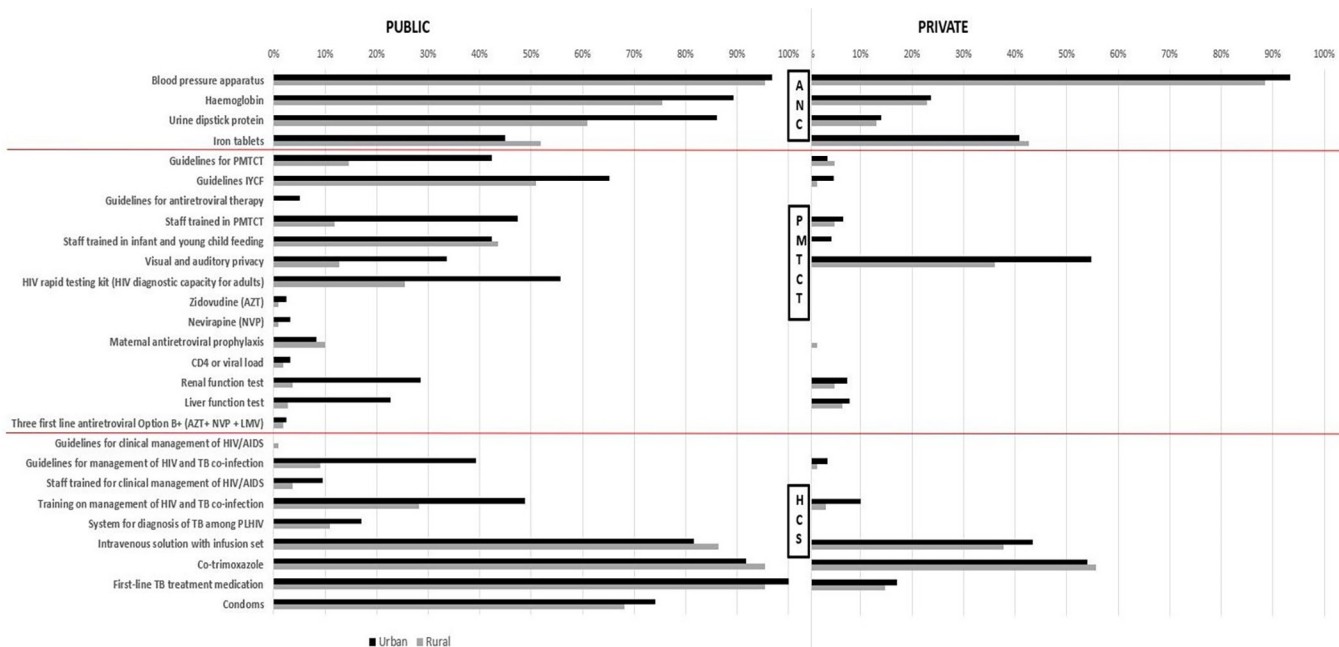

**Fig 2. ANC, PMTCT and HIV Care and Support (HCS) indicators, by urban/rural and public/private, 2016. Note:** Four variables in ANC service indicators are excluded from this figure since the comparable variable are not available for private facilities–these are guidelines and staff trained in ANC, and the availability of folic acid tablets and tetanus toxoid vaccine. AIDS: Autoimmune Deficiency Syndrome; ANC: Antenatal Care; AZT: Zidovudine; CD4: Cluster of Differentiation 4; HIV: Human Immunodeficiency Virus; IYCF: Infant Young and Child Feeding; LMV: Lamovudine; PLHIV: People Living with Human Immunodeficiency Virus; PMTCT: Prevention of Mother-to-Child Transmission; NVP: Nevirapine; TB: Tuberculosis.

## Univariate and multivariate results

Table 2 shows that based on the univariate analysis, the proportion of pregnant women tested for HIV was more likely to be greater than or equal to 10% in facilities with a higher readiness score, facilities connected to a larger number of community health posts, and facilities where there were more trained counsellors available onsite. This result was consistent across each of the models for ANC, PMTCT and HCS. The multivariate analysis shows that the proportion of pregnant women tested for HIV was more likely to be greater than or equal to 10% in facilities with a higher readiness score, a higher number of trained counsellors available, and less likely in facilities located outside of Java-Bali or in facilities supporting a higher number of

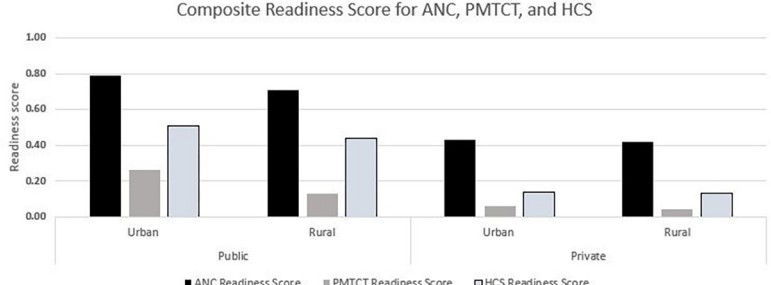

**Fig 3. Composite readiness scores for ANC, PMTCT and HCS, by urban/rural and public/private, 2016. Note:** ANC: antenatal care; PMTCT: Prevention of Mother-to-Child Transmission; HCS: HIV care and support. Four variables in ANC service indicators were excluded since the comparable variables were not available for private facilities–these are guidelines and staff trained in ANC, and the availability of folic acid tablets and tetanus toxoid vaccine.

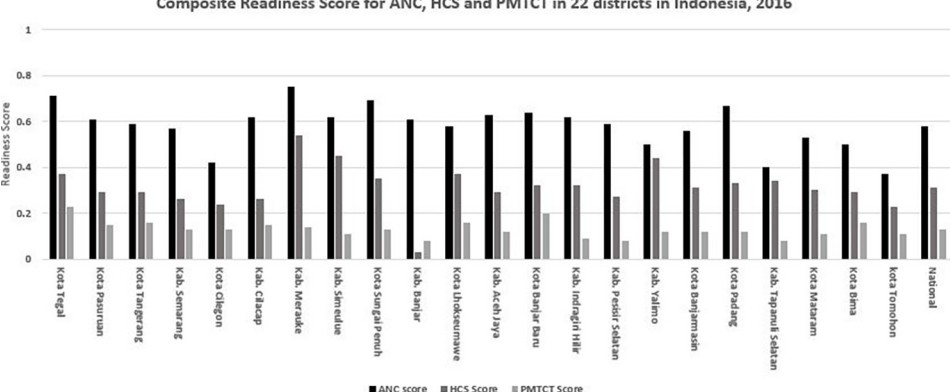

**Fig 4. Composite readiness scores for ANC, PMTCT and HCS, by districts, 2016. Note:** ANC: antenatal care; PMTCT: Prevention of Mother-to-Child Transmission; HCS: HIV care and support. Four variables in ANC service indicators are excluded from calculation since the comparable variable are not available for private facilities–these are guidelines and staff trained in ANC, and the availability of folic acid tablets and tetanus toxoid vaccine.

village midwives. Results of the sensitivity analysis show that varying the threshold for 10% did not have a significant impact on the models (see S2 and S3 Tables).

Despite an increase in the national antenatal HIV testing coverage from 10% in 2016 to 34% in 2018, the changes were not uniform across facilities. From 2016 to 2018, around 20% of the facilities decreased between 1% to 37%, 33% increased between 0.1% to 25%, and 47% of the facilities increased by more than 25%. Our multivariate regression analysis shows that none of the readiness indicators were significantly associated with the rate of absolute change in antenatal HIV testing coverage per facility (see S4 Table).

## Discussion

Despite the Government of Indonesia's commitment to the elimination of mother-to-child transmission of HIV, syphilis, and hepatitis B by 2022, testing coverage for HIV remains low and subsequent treatment uptake even lower. While the national PMTCT policy was adopted in 2005 in Indonesia using voluntary counselling and testing approach, it was only in 2017 that the national guidelines introduced HIV testing as a part of routine testing in antenatal care settings [13, 15]. Gaps in supply-side readiness, such as the availability of rapid testing kits and treatment, has been posited as a potential determinant of this lack of uptake [20, 46, 52]. This analysis has confirmed that, in 2016, widespread gaps existed in the infrastructure, equipment, diagnostics, and human resources that are necessary for the delivery of PMTCT services for HIV in Indonesia. It also confirms that these gaps are a significant predictor of poor uptake of testing for HIV amongst pregnant women.

Key gaps in supply-side readiness for the delivery of PMTCT for HIV in Indonesia included shortages in the supply of rapid tests and treatments, trained staff including experienced counsellors, the provision of PMTCT guidelines, and private space for consultations and counselling. Our finding that facilities in rural areas are most lacking is consistent with several other LMICs studies [53–55]. Major disparities in readiness scores across districts in Indonesia also align with a broader literature highlighting differences in the quality of care at the subnational level in LMICs, especially in the delivery of obstetric services and primary care [56, 57].

While private providers are the first port of call for around half of all pregnant women in Indonesia [28], this study has shown that they are lacking even the most basic of inputs needed to deliver PMTCT services. For example, less than 1% of private facilities maintained adequate

**Table 2. Results of multivariate logistic regression model of factors associated with the proportion of ≥10% of pregnant women tested for HIV per facility in public facilities, in 2016 in Indonesia.**

| Variables | | Model 1: ANC Readiness Scores | | | | Model 2: PMTCT Readiness Scores | | | | Model 3: HCS Readiness Scores | | | |
|---|---|---|---|---|---|---|---|---|---|---|---|---|---|
| | | uOR (95% CI) | p | aOR (95% CI) | p | uOR (95% CI) | p | aOR (95% CI) | p | uOR (95% CI) | p | aOR (95% CI) | p |
| Readiness scores[1] | 0 | 1.00 | | 1.00 | | 1.00 | | 1.00 | | 1.00 | | 1.00 | |
| | 1 | 2.89 (0.70 to 11.97) | 0.144 | 4.41 (0.62 to 31.25) | 0.138 | 4.89 (1.86 to 12.84) ** | 0.001 | 1.11 (0.17 to 7.12) | 0.915 | 1.96 (0.61 to 6.29) | 0.256 | 3.35 (0.60 to 18.73) | 0.169 |
| | 2 | 10.11 (2.97 to 34.36)** | <0.001 | 7.39 (1.55 to 35.25)** | 0.012 | 33.37 (7.48 to 148.39) ** | <0.001 | 14.95 (2.85 to 78.42)** | 0.001 | 3.74 (1.23 to 11.33)** | 0.020 | 3.61 (0.61 to 21.30) | 0.156 |
| Regions | Java-Bali | 1.00 | | 1.00 | | 1.00 | | 1.00 | | 1.00 | | 1.00 | |
| | Outer Java-Bali | 0.01 (0.003 to 0.024)** | <0.001 | 0.04 (0.01 to 0.21)** | <0.001 | 0.01 (0.003 to 0.024) ** | <0.001 | 0.02 (0.003 to 0.102)** | <0.001 | 0.01 (0.003 to 0.024)** | <0.001 | 0.02 (0.003 to 0.126)** | <0.001 |
| Areas | Urban | 1.00 | | 1.00 | | 1.00 | | 1.00 | | 1.00 | | 1.00 | |
| | Rural | 0.15 (0.06 to 0.37)** | <0.001 | 1.10 (0.22 to 5.46) | 0.905 | 0.15 (0.06 to 0.37)** | <0.001 | 1.09 (0.15 to 7.73) | 0.929 | 0.15 (0.06 to 0.37)** | <0.001 | 1.29 (0.32 to 5.10) | 0.718 |
| Type of services | BEONC[2] | 1.00 | | | | 1.00 | | | | 1.00 | | | |
| | Non-BEONC[2] | 0.59 (0.24 to 1.45) | 0.253 | | | 0.59 (0.24 to 1.45) | 0.253 | | | 0.59 (0.24 to 1.45) | 0.253 | | |
| Type of Financial Managements | BLUD[3] | 1.00 | | | | 1.00 | | | | 1.00 | | | |
| | Non-BLUD[3] | 1.02 (0.36 to 2.91) | 0.973 | | | 1.02 (0.36 to 2.91) | 0.973 | | | 1.02 (0.36 to 2.91) | 0.973 | | |
| Number of village midwives | | 0.87 (0.82 to 0.93)** | <0.001 | 0.93 (0.80 to 1.08) | 0.365 | 0.87 (0.82 to 0.93)** | <0.001 | 0.93 (0.80 to 1.07) | 0.318 | 0.87 (0.82 to 0.93)** | <0.001 | 0.93 (0.81 to 1.08) | 0.358 |
| Number of trained counsellors | | 3.06 (1.89 to 4.96)** | <0.001 | 1.73 (1.06 to 2.80)** | 0.026 | 3.01 (1.89 to 4.96)** | <0.001 | 1.78 (1.14 to 2.79)** | 0.011 | 3.01 (1.89 to 4.96)** | <0.001 | 1.70 (1.07 to 2.67)** | 0.022 |
| Number of active CHWs | | 1.01 (1.00 to 1.02)** | <0.001 | 1.00 (0.99 to 1.00)* | 0.083 | 1.01 (1.00 to 1.01)** | <0.001 | 1.00 (0.99 to 1.01) | 0.292 | 1.01 (1.00 to 1.01)** | <0.001 | 1.00 (0.99 to 1.01) | 0.173 |
| Number of community health posts | | 1.05 (1.02 to 1.08)** | <0.001 | 1.00 (0.95 to 1.05) | 0.899 | 1.05 (1.02 to 1.08)** | <0.001 | 0.98 (0.94 to 1.03) | 0.510 | 1.05 (1.02 to 1.08)** | <0.001 | 0.99 (0.94 to 1.04) | 0.739 |

NOTES

**p-value: <0.05

*p-value: <0.1

ANC readiness score: 0 = 0–0.65; 1 = 0.66–0.80; 2 = 0.81–1; PMTCT readiness score: 0 = 0–0.22; 1 = 0.23–0.40; 2 = 0.41–1; HCS readiness score: 0 = 0–0.35; 1 = 0.36–0.50; 2 = 0.51–1.

ANC: antenatal care; aOR: adjusted odds ratio; CHWs: community health workers; CI: confidence interval; HCS: HIV care and support; PMTCT: Prevention Mother-to-Child Transmission; uOR: unadjusted odds ratio

[1]Model 1 used ANC readiness scores, model 2 used PMTCT readiness scores and model 3 used HCS readiness scores

[2]BEONC: Basic Emergency, Obstetric and Neonatal Care, referring to the health facilities ("*Puskesmas*") that were equipped with the capacity to provide basic obstetric and neonatal emergency care

[3]"*Badan Layanan Umum Daerah/BLUD*" is a term used for a public district organization that provides services to the community with the flexibility to implement business models to support revenue generation and to improve efficiency

supplies of testing kits and treatments for HIV at the time of the QSDS survey, compared to around 10% in the public sector. Evidence on the readiness of private providers to deliver primary care in LMICs is mixed. For example, a recent study found that public providers were better prepared than their private counterparts to deliver intermittent presumptive treatment for malaria in Tanzania [22]. In contrast, studies in Benin [58] and in Nepal [59] have reported higher readiness scores for the delivery of maternal care and general health services in private facilities compared to public ones. While data on HIV testing and treatment in pregnancy from private providers in Indonesia has been demonstrated to be lacking and the health system

between public and private sectors was fragmented [15, 35], it is clear that scaling up PMTCT services in Indonesia will be implausible without strengthening the private sector [17]. There is a need to extend and strengthen the national surveillance system for HIV/AIDS to incorporate data on testing and treatment for HIV among pregnant women in the private sector.

There is a dearth of research on supply-side determinants of the uptake of antenatal HIV testing in LMICs [25, 60]. Our analysis begins to address this gap by firstly demonstrating that pregnant women attending public facilities with higher service readiness scores are more likely to be tested for HIV. This supports the widely held view that delivering and monitoring health care inputs is a necessary prerequisite to achieving universal coverage [15, 48, 61]. Studies of antenatal care in Nigeria [62], child health care in Malawi [54], obstetric care in Haiti [57] and multi-country analyses of obstetric services [56] all highlight positive associations between supply-side readiness and the uptake of these services. We also found that testing uptake was less likely in facilities located in areas with a higher number of village midwives. This finding could reflect a lack of capacity among village midwives to recognise the need to refer women to health centres or hospitals (including for HIV testing) and to manage complicated births [63, 64]. Village midwives are one of the main providers of maternal care in Indonesia [28] and expansion of the PMTCT services requires strong engagement and clear linkages between these providers and other health facilities for HIV care, support and treatment services [15, 65]. Lastly, our study showed that testing was more likely to occur in facilities located in urban districts or on the main island of Java-Bali. PMTCT activities can put additional pressure on rural facilities, many of which are already under-staffed and under-resourced [24]. Reported delays and barriers to the transfer of funds from the central government to support poorer districts to carry out PMTCT services need to be addressed [15]. In terms of implementation, it is ultimately the district health authorities that have the legal responsibility to provide health care in Indonesia according to their own priorities [66]. Ensuring these priorities align with the national PMTCT policy and that there are adequate funds to support its delivery to around five million pregnant women every year is an ongoing challenge for the Indonesian government.

A few limitations of this study need to be acknowledged. First, we were unable to report on any improvements of supply-side readiness for the delivery of PMTCT of HIV in Indonesia beyond 2016. However, recent policy debates suggest that weak service readiness, especially among private maternal health providers is an ongoing challenge for Indonesia [1, 27, 67]. Our findings serve as a useful baseline measure for future evaluations of interventions to strengthen inputs for PMTCT. Qualitative information gathered by the WHO national review of HIV services in Indonesia also indicated that HIV testing and access to ART for pregnant women is very low in the private sector [15]. Second, due to the scarcity of surveillance data on the HIV cascade of care in the ANC setting, we were unable to explore associations between supply-side readiness and the number of pregnant women receiving treatment for HIV. Third, the SIHA database system is not designed for research purposes and hence, several relevant variables for the analysis of PMTCT were not recorded, including the number of pregnant women attending their first antenatal care visit, the ratio of pregnant women to health workers, and any external/foreign financial support received by facilities for PMTCT. Lastly, this study is based on an ecological study design, therefore we cannot confirm causal relationships between supply-side readiness and HIV testing uptake.

## Conclusion

While it is recognised that improving the availability of resources in facilities for the provision of services like PMTCT will not on its own lead to major improvements in the quality of care,

the availability of key inputs such as infrastructure, equipment, diagnostics, and human resources does provide a critical foundation for the delivery of care. This is especially the case for services such as PMTCT, which rely heavily on specific service inputs such as rapid tests and essential medicines. If Indonesia's target for the triple elimination of mother-to-child transmission of HIV, syphilis and hepatitis B is to be achieved by 2022, strengthening the capacity of all primary health care providers to deliver PMTCT services must be a priority. This includes better integration of private providers, especially private midwives, into the Indonesian health care system.

## Supporting information

**S1 Table. List of variables for logistic regression model.**
(DOCX)

**S2 Table. Multivariate logistic regression model with 5% threshold.**
(DOCX)

**S3 Table. Multivariate logistic regression model with 15% threshold.**
(DOCX)

**S4 Table. Regression model with the rate of absolute change in antenatal HIV testing from 2016 to 2018.**
(DOCX)

**S5 Table. Modified PMTCT readiness score with one indicator excluded.**
(DOCX)

**S6 Table. Results of multivariate logistic regression model of factors associated with the proportion of $\geq$10% of pregnant women tested for HIV per facility in public facilities, in 2016 in Indonesia–with modified PMTCT readiness score.**
(DOCX)

## Acknowledgments

The authors thank the Sub-Directorate HIV/AIDS and STIs of the Indonesian MoH, especially Indrawati Victoria and Sugeng Wiyana, for supporting the res earch and assisting with the PMTCT data extraction from the national System Information for HIV AIDS (SIHA).

## Author Contributions

**Conceptualization:** Rabiah al Adawiyah, Tanya L. Applegate, Ari Probandari, Tiara Marthias, Rebecca Guy, Virginia Wiseman.

**Data curation:** Rabiah al Adawiyah, David Boettiger, Tiara Marthias.

**Formal analysis:** Rabiah al Adawiyah, David Boettiger, Tanya L. Applegate, Ari Probandari, Virginia Wiseman.

**Methodology:** Rabiah al Adawiyah.

**Writing – original draft:** Rabiah al Adawiyah, David Boettiger.

**Writing – review & editing:** Rabiah al Adawiyah, David Boettiger, Tanya L. Applegate, Ari Probandari, Tiara Marthias, Rebecca Guy, Virginia Wiseman.

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
