## [Decision Letter · Decision Letter 0]

27 May 2022

PGPH-D-21-00735

Supply-side readiness to deliver HIV testing and treatment services in Indonesia: Going the last mile to eliminate mother-to-child transmission of HIV

Dear Dr. Adawiyah,

Thank you for submitting your manuscript to PLOS Global Public Health. After careful consideration, we feel that it has merit but does not fully meet PLOS Global Public Health’s publication criteria as it currently stands. Therefore, we invite you to submit a revised version of the manuscript that addresses the points raised during the review process.

We look forward to receiving your revised manuscript.

Kind regards,

Sanghyuk Shin

Academic Editor

Journal Requirements:

1. In your ethics statement in the Methods section and in the online submission form, please confirm whether all data were fully anonymized before you accessed them.

2. Please update the 'Competing Interests' statement in the system, including any COIs declared by your co-authors. If you have no competing interests to declare, please state "The authors have declared that no competing interests exist". 

3. Please amend your detailed Financial Disclosure statement. This is published with the article. It must therefore be completed in full sentences and contain the exact wording you wish to be published.

- State what role the funders took in the study. If the funders had no role in your study, please state: “The funders had no role in study design, data collection and analysis, decision to publish, or preparation of the manuscript.”

4. Please provide separate figure files in .tif or .eps format and remove them from the manuscript file.

5. We have noticed that you have uploaded Supporting Information files, but you have not included a list of legends. Please add a full list of legends for your Supporting Information files after the references list. 

Additional Editor Comments (if provided):

Thank you for your patience. Please  address comments from Reviewer #1, with emphasis on explanation of how private midwifes fit into your analysis. In addition, there are several issues with the model results presented in Table 2 that should be addressed:

- The Readiness Scores presented in Table 2 do not seem to align with the explanation provided in the Methods whereby each score should be a proportion. Please explain the definition and meaning of the 0, 1, 2 readiness scores presented in Table 2 in the Methods and in a footnote for Table 2.

- In the description of Table 2 in the Results (page 14), higher readiness score is associated with the outcome in univariate analysis. As these are the primary exposure variables, the associations for these variables in the multivariable model should also be described in Results.

- Page 9 in Methods describes the outcome variable as HIV testing among pregnant women >= 10%. Why was this threshold used? If it’s a standard threshold, please explain and provide citation.

- Please explain in the Methods why you chose to fit three separate models for each readiness score exposure variable, instead of including all three in one model.

Please address minor issues below:

- In the Introduction of the Abstract, it would be helpful to briefly define readiness.

- In the Abstract, multivariable analysis is mentioned in the Methods, but the Results section does not include findings from the multivariable analysis. Please add a brief summary of the findings in Results.

- Page 9. Cronbach’s alpha for PMTCT readiness scores is low at 0.630. Please describe which items are responsible for the low alpha. A sensitivity analysis with a modified PMTCT readiness score with the items removed might be useful.

- In page 11, the readiness scores for each three types of services are compared. Since the number of items and the content of the measures are different, it seems in appropriate to compare the scores across these different services. Please explain in Methods why this comparison is appropriate or remove the comparison from the paper.

- References 31 & 34 are duplicates. Please eliminate one of them.

- Some of the references seem to be missing information, including journal name, date, pages. For example, references 51 and 59, Please go through all of the references and make sure they have complete information in the correct format.

Reviewers' comments:

Reviewer's Responses to Questions

**Comments to the Author**

1. Does this manuscript meet PLOS Global Public Health’s publication criteria? Is the manuscript technically sound, and do the data support the conclusions? The manuscript must describe methodologically and ethically rigorous research with conclusions that are appropriately drawn based on the data presented.

Reviewer #1: Yes

2. Has the statistical analysis been performed appropriately and rigorously?

Reviewer #1: I don't know

3. Have the authors made all data underlying the findings in their manuscript fully available (please refer to the Data Availability Statement at the start of the manuscript PDF file)?

Reviewer #1: Yes

4. Is the manuscript presented in an intelligible fashion and written in standard English?

Reviewer #1: Yes

5. Review Comments to the Author

Reviewer #1: The paper addresses an international public health priority, PMTCT. Such findings contribute to the question of “how to reach elimination”.

For info, recent paper from Indonesia, on the same topic : Doi : 10.1177/23259582211040701

The authors use large surveys/databases, routine data. They used different frameworks to guide their analysis (for barriers to access to care, for health systems readiness), which increases the quality and replicability of their work.

The description of the health system and MCH offer is quite complex, maybe a figure/map would help.

Authors explain that many/most pregnant women seek care from private midwife clinics; but also explain that village midwives provide most MHC care – are these based in the private clinics? What are their links? The study sample include private facilities but does it include these private midwife clinics?

Is there a readiness threshold? above which the components of the health system are considered as "ready"?

The Indonesia PMTCT policy context seems to be specific. The integration of HIV testing in routine ANC in 2017 seems late compared to other countries (opt-out approach to testing, PICT integrated in mist African countries since early/mid 2000s). What are the reasons for that. Time between policy, political decision, funding decision and intervention implementation can be "quite long", this should be mentioned in the discussion to properly interpret findings.

6. PLOS authors have the option to publish the peer review history of their article (what does this mean?). If published, this will include your full peer review and any attached files.

**Do you want your identity to be public for this peer review?** For information about this choice, including consent withdrawal, please see our Privacy Policy.

Reviewer #1: No

---

## [Editor Report · Decision Letter 1]

7 Jul 2022

Supply-side readiness to deliver HIV testing and treatment services in Indonesia: Going the last mile to eliminate mother-to-child transmission of HIV

PGPH-D-21-00735R1

Dear Dr Adawiyah,

We are pleased to inform you that your manuscript 'Supply-side readiness to deliver HIV testing and treatment services in Indonesia: Going the last mile to eliminate mother-to-child transmission of HIV' has been provisionally accepted for publication in PLOS Global Public Health.

Best regards,

Sanghyuk Shin

Academic Editor

Thank you for thoroughly and very clearly addressing all of the reviewer and editor's comments.